# Towards Effective Model Editing for LLM Personalization

## Abstract

Customer-facing LLMs need personalization to reflect individual preferences and needs. However, existing personalization methods are often computationally expensive, data-intensive, prone to catastrophic forgetting, and degrade in multi-turn conversations and on implicit questions. To address these challenges, we conceptualize personalization as *model editing* and present ***Personalization Editing***, a framework that applies localized edits guided by clustered preference representations, enforcing desired behavior where preferences apply while preserving other capabilities. Existing personalization datasets often use synthetic personas in role-playing dialogues, leading to indirect evaluation that does not reflect real-world user queries. We introduce **UPQA** , a short-answer QA dataset based on in-situ user queries, with varying levels of difficulty. Unlike prior benchmarks, UPQA directly tests whether models can recall and apply specific user preferences, enabling more accurate and efficient evaluation. Across settings, Personalization Editing improves editing accuracy, and is more computationally efficient than fine tuning, while outperforming prompting and retrieval based baselines in multi-turn conversations and on implicit preference questions. [1]

## 1 Introduction

Large Language Models (LLMs) deliver state-of-the-art performance across core Natural Language Processing (NLP) tasks such as text generation, translation, question answering, and chatbots (Brown et al., 2020; Thoppilan et al., 2022). Beyond benchmarks, there is growing interest in tailoring LLMs to individual users. Personalization tailors model outputs on preferences, goals, and context derived from interaction history and other user signals, improving relevance and user satisfaction. Its potential has driven applications in e-commerce (Geng et al., 2022; Li et al., 2024), education (Wang et al., 2024a; Stamper et al., 2024), and healthcare (Singhal et al., 2023; 2025).

Personalization of LLMs introduces unique challenges (Salemi et al., 2023). Methods that rely on additional domain specific data and extensive fine tuning are computationally costly, data intensive, and prone to catastrophic forgetting, with increased business risk and user dissatisfaction (Laban et al., 2025; Liu et al., 2023; Zhang et al., 2024b). In-context approaches such as prompt engineering and Retrieval-Augmented Generation (RAG) inject user profile features at inference time, but performance degrades in multi-turn conversations as key information is diluted, and these methods struggle on implicit questions that require reasoning beyond explicit user profile facts (Bai et al., 2024; Zhao et al., 2025). These limitations motivate methods that reduce compute and data demands while mitigating catastrophic forgetting, especially in multi-turn conversations.

Model editing, also known as knowledge editing, makes precise, targeted changes to LLM behavior with minimal data and computation (Wang et al., 2024b). It can be used as a parameter-efficient route to LLM personalization, to avoid the substantial costs of full fine-tuning and mitigate catastrophic forgetting in multi-turn conversations (Huang et al., 2025). In particular, we take personalization as a constrained model-editing problem: for each user preference, we replace the default response for a given subject–predicate with the user-preferred one, and learn an editing model that produces the desired outputs on those inputs while leaving all other behaviors unchanged. Two challenges arise. On the one hand, edits must be local, changing behavior only when a user preference is relevant

---

[1]Code, data, and results are available at https://anonymous.4open.science/r/personalization-model-editing

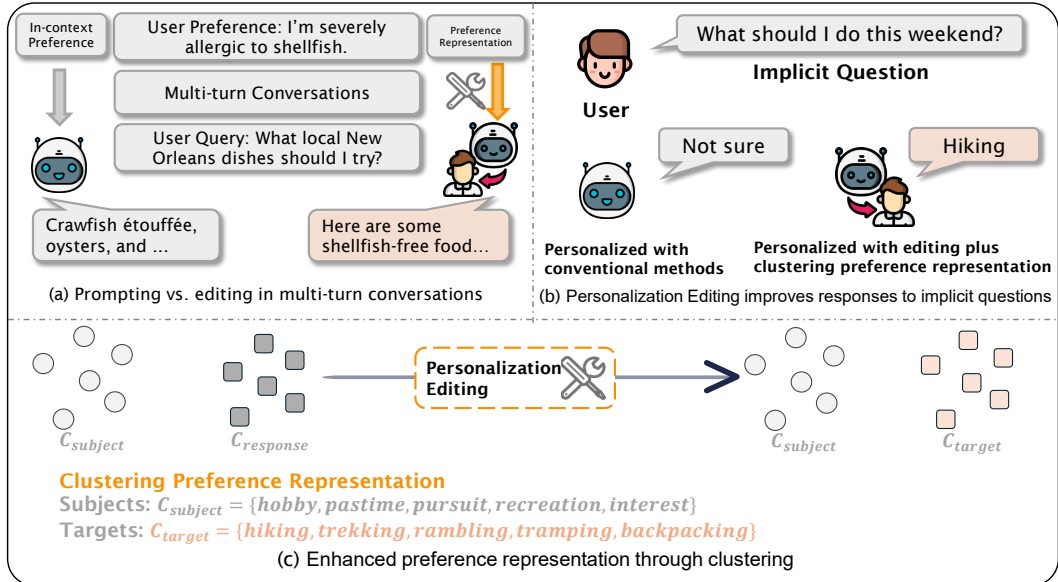

Figure 1: The proposed Personalization Editing framework outperforms prompting-based methods in multi-turn conversations and surpasses fine-tuning-based approaches in both accuracy and efficiency. Furthermore, its enhanced clustering-based preference representation enables the framework to recall user preferences even in challenging implicit queries, where existing methods often fail.

while preserving responses to unrelated queries. On the other hand, representing user preferences as a fixed response may fail to capture semantically related intents. For example, for a user who enjoys outdoor activities, the question "What should I do this weekend?" should admit a series of similar concepts such as hiking, trekking, rambling, and tramping, rather than a single canonical answer.

To address these issues, we propose a novel framework *Personalization Editing*, a framework that uses clustering-based preference representations to improve the robustness of edits. Instead of a single fixed response, preferences are represented by clusters that capture groups of semantically similar keywords (subjects) and their corresponding targets. This design sustains strong performance in multi-turn conversations, where in context methods degrade, and it handles real implicit preference questions in which users express needs indirectly.

Moreover, most existing personalization datasets focus on persona based dialogue in fictional, open domain settings (Jandaghi et al., 2023). These are useful for studying conversational style but do not test whether a model can faithfully integrate real user information. We therefore introduce UPQA (Under Preference Question Answering), a benchmark that evaluates whether models recall and apply user profile facts. UPQA includes structured questions for explicit knowledge, probes for implicit preferences, and practical scenarios such as product recommendations.

Our framework improves editing accuracy, robustness in multi-turn conversations, and computational efficiency over strong baselines. Our key contributions are as follows:

- **Model editing for personalized LLM.** We study a novel problem by leveraging model editing to personalize LLMs to enable precise model updates without requiring expensive retraining, while preserving the model's general capabilities. It also outperforms prompting-based personalization techniques in multi-turn conversations.
- **Clustering-based preference representation for robust, accurate model editing.** The proposed method uses a clustering-based preference representation that augments existing model editing techniques, which improves the robustness and accuracy of these methods when handling challenging implicit preference questions across diverse domains.
- **UPQA: a multi-domain personalization editing dataset for standardized evaluation.** We introduce UPQA, a challenging dataset designed for rigorous and standardized evaluation of personalization editing across diverse scenarios. By adopting a short-answer QA format with varying levels of difficulty, UPQA enables efficient and reliable assessment of parameter-based personalization methods.

## 2 RELATED WORK

In this section, we briefly review two related topics: LLM personalization and model editing.

### 2.1 LLM PERSONALIZATION

LLM personalization adapts models to individual user needs and preferences, enhancing satisfaction through more relevant interactions (Zhang et al., 2024b). In-context approaches, such as profile-augmented prompts (injecting a user's profile or history into the prompt) (Zhang et al., 2018) or RAG (Fan et al., 2024) that fetches user-specific information from an external memory, incorporate user data into the model's input without altering the model's weights. However, compressing a user's history into a prompt can cause information loss, and they rely on limited context windows (Liu et al., 2025). Fine-tuning approaches instead update the model's parameters on user data – for example, training a personalized adapter module within the model's layers (Zhong et al., 2021) – or using Reinforcement Learning from Human Feedback (RLHF) to align the model with human preferences (Ouyang et al., 2022). However, they are often resource-intensive and prone to overfitting (Liu et al., 2025). Moreover, reward-alignment techniques like Bai et al. (2022); Rafailov et al. (2023) typically optimize for generic user preferences, so capturing each individual's unique style or values would require costly user-specific feedback data (Liu et al., 2025). Hence in this paper, we mainly focus on parameter-efficient fine-tuning strategies such as Low-Rank Adaptation (LoRA) (Hu et al., 2022) and prefix-tuning (Vos et al., 2022) to personalize LLMs, which update only a small fraction of the model's parameters while keeping most weights frozen.

### 2.2 MODEL EDITING

Model editing enables efficient and precise modification in LLMs without full retraining and finetuning while largely preserving nontargeted capabilities (Wang et al., 2024b). Editing methods fall into three categories: locate-then-edit approaches like ROME (Meng et al., 2022) that identify and modify specific factual associations, and its multi-edit successor MEMIT (Meng et al., 2023); parameter-efficient fine-tuning methods that update targeted parameters using techniques like fine-tuning with masking (FT-M) (Rozner et al., 2024; Gangadhar & Stratos, 2024); and in-context editing like IKE (Zheng et al., 2023a) that embeds corrections directly in input context. These techniques effectively update outdated knowledge and reduce hallucinations while preserving model capabilities. By leveraging such model editing techniques, we aim to achieve efficient personalization of LLMs, integrating user-specific knowledge or preferences into the model on the fly without the full costs of retraining or the context limitations of prompting. However, existing methods are not suitable for modeling semantically related intents in user-preference–based recommendation, as mentioned in the introduction. To address this gap, we introduce a clustering-based preference representation method that augments model editing techniques, improving their robustness and accuracy on challenging implicit preference questions.

## 3 PROBLEM FORMULATION

### 3.1 PERSONALIZATION AS EDITING

The goal of Personalization Editing is to precisely and efficiently adapt an LLM to reflect individual user needs, preferences, and behaviors, while preserving its general capabilities across the broader user population. Personalization Editing operates on a structure analogous to a knowledge tuple $(s, r, o)$, where $s$ represents a subject, $r$ denotes predicate, and $o$ represents object.

In the personalization setting, the goal is to modify model responses to align with specific user preferences in given contexts. This is formalized as transforming an original tuple $(s, r, o)$ into a new tuple $(s, r, o^*)$ that reflects the personalized preference, where $o^*$ represents the desired personalized response or behavior preference. Here, the user and predicate remain constant while the response adapts to user-specific preferences.

**Example.** If a user profile records hobbies *running* and *reading*, tuples include $(\text{user}, \text{hobby}, \text{running})$ and $(\text{user}, \text{hobby}, \text{reading})$. After editing, when asked "Suggest some

activities for the weekend," the model recommends a local trail run or a book club rather than generic activities.

## 3.2 INPUT–OUTPUT MAPPING

To probe and modify model responses for personalization, the subject $s$ must be converted into a natural language question $x$, to which the model responds with an output $y$. This input-output pair is associated with a tuple $(s, r, o)$. The input space corresponding to a personalization edit is denoted as $\mathcal{X}_e = I(s, r)$, where $I$, where $I$ is a question-generation function that maps the subject and relation to a set of relevant input questions. The original output space is defined as $\mathcal{Y}_e = O(s, r, o)$, and the desired personalized output space after editing is represented as $\mathcal{Y}_e^* = O^*(s, r, o^*)$. For a single edit $e$ with input space $\mathcal{X}_e$, the objective of Personalization Editing is to transform the original outputs $\mathcal{Y}_e$ into the target outputs $\mathcal{Y}_e^*$.

## 3.3 MULTIPLE EDITS

When considering a set of personalization edits $\mathcal{E} = \{e_1, e_2, \ldots\}$, the combined input space is $\mathcal{X}_{\mathcal{E}} = \bigcup_{e \in \mathcal{E}} \mathcal{X}_e$, and the corresponding original and target output spaces are $\mathcal{Y}_{\mathcal{E}} = \bigcup_{e \in \mathcal{E}} \mathcal{Y}_e$ and $\mathcal{Y}_{\mathcal{E}}^* = \bigcup_{e \in \mathcal{E}} \mathcal{Y}_e^*$, respectively.

## 3.4 OBJECTIVE

Let the original LLM be a function $f : \mathcal{X} \to \mathcal{Y}$. The goal of Personalization Editing is to produce a personalized model $f^* : \mathcal{X} \to \mathcal{Y}^*$, such that the edited model generates personalized outputs for inputs in $\mathcal{X}_{\mathcal{E}}$ while preserving its responses on all other inputs, preventing degradation of unrelated model behavior. The optimization aims to minimize the discrepancy between the personalized output $f^*(x)$ and the desired target output $y^*$, as measured by a loss function $\mathcal{L}$. Simultaneously, the editing must maintain consistency on all inputs outside the editing set, ensuring that $f^*(x) = f(x)$ for all $x \in \mathcal{X} \setminus \mathcal{X}_{\mathcal{E}}$. This yields the constrained optimization objective:

$$\min \mathbb{E}_{e \in \mathcal{E}} \mathbb{E}_{x, y^* \in \mathcal{X}_e, \mathcal{Y}_e^*} \mathcal{L}(f^*(x), y^*)$$
$$\text{s.t. } f^*(x) = f(x), \quad \forall x \in \mathcal{X} \setminus \mathcal{X}_{\mathcal{E}}$$

# 4 DATASET CONSTRUCTION

To rigorously evaluate personalization editing, we consider two datasets. First, we introduce UPQA , a short-answer QA benchmark built from in-situ user queries, specifically designed for standardized and efficient evaluation of personalization editing. Second, we adapt PREFEVAL (Zhao et al., 2025), a multi-turn conversation benchmark, to better align with model editing. Together, these datasets provide a diverse and challenging testbed for assessing both the accuracy and robustness of personalization methods.

## 4.1 UPQA (USER PREFERENCE QUESTION ANSWERING)

We curated **UPQA** by extracting user-profile features from the *Synthetic Persona Chat* dataset (Jandaghi et al., 2023). We first aggregated all unique persona attributes, where each attribute encodes a specific user preference (e.g., "I enjoy hiking," "I have a dog," "I work as a teacher"). These serve as the foundation for personalization evaluation.

To transform persona attributes into structured evaluation queries, we employed `Claude-Sonnet-4`, selected for its strong performance on instruction-following benchmarks (Sharma et al., 2025). The model was prompted to analyze each persona attribute and generate a suite of *in-situ* user queries at different levels of difficulty. This process ensured that evaluation questions are both naturalistic and systematically varied.

For each user preference, we annotated an `attribute_type`, a high-level category of personal information such as *hobby, profession, family, pet, or location*. Based on this categorization, we designed four complementary query types:

1. `question`: Direct queries about the attribute using its exact term (e.g., "What's my hobby?" for a hiking-related persona).
2. `question_paraphrased`: Rephrasings of the direct query in more natural language.
3. `implicit_question`: Indirect queries that rely on prior knowledge (e.g., "What should I do this weekend?" for a hiking hobby).
4. `product_recommendation_question`: Preference-grounded recommendation requests that omit explicit attribute values (e.g., "Any gear I should buy for my hobby?").

We collected over 1,000 unique user preferences spanning a broad range of topics, including hobbies, family roles, professions, and personal attributes. To ensure fairness in evaluation, we sampled a balanced subset with equal representation across attribute types. This prevents bias toward over-represented categories and supports standardized comparison across methods.

To support cluster-based editing and representation learning, we further augmented UPQA by generating clusters of nine semantically related synonyms for each editing subject and target. This augmentation captures lexical variability (e.g., "dog," "puppy," "canine") while preserving semantic meaning, ensuring robustness in both evaluation and model editing. Synonym clusters were again generated with `Claude-Sonnet-4`, followed by manual verification.

Below is an example from UPQA . An example of augmented data, including attribute annotations, question types, and synonym clusters, is provided in the Appendix E.

```
{
    "attribute_type": "food preference",
    "question": "What's my food preference?",
    "question_paraphrased": "What kind of food do I like best?",
    "implicit_question": "What should I have for dessert tonight?",
    "target": "Any products I should buy for my food preference?",
    "input_attribute": "My favorite food is ice cream."
}
```

## 4.2 PREFEVAL

In addition to UPQA , we also evaluate our framework on the PREFEVAL benchmark (Zhao et al., 2025), where user preferences are expressed directly in single-turn utterances, and subsequent queries test whether models can recall and apply these preferences after long, noisy contexts. PREFEVAL serves as a baseline for assessing preference following; however, it is primarily designed for prompting and retrieval-based methods rather than model editing.

To adapt PREFEVAL for our setting, we reformulate the benchmark into structured key–value pairs by extracting `subject` and `target`. This restructuring isolates the core preference signal and enables precise updates, facilitating efficient preference injection or correction without retraining on entire conversations. The prompt used with `Claude-Sonnet-4` for augmentation is provided in the Appendix C. An example of the augmented data is given below.

```
{
    "topic": "travel_restaurant",
    "preference": "I have a severe peanut allergy, so I must avoid any foods containing
    peanuts or peanut products.",
    "question": "I'm visiting Thailand next month. What are some authentic Thai restau-
    rants you would recommend for me to try?",
    "explanation": "Thai cuisine commonly utilizes peanuts and peanut-based sauces, so
    recommending authentic Thai restaurants presents a challenge given the user's peanut al-
    lergy.",
    "subject": "restaurants",
    "target": "peanut-free"
}
```

## 5 EXPERIMENTS

### 5.1 BASELINE METHODS

- FT-L (Meng et al., 2022) Constrained fine-tuning that targets a specific FFN layer identified by causal tracing, maximizing likelihood of target sequences with parameter-space norm constraints to minimize interference with unmodified facts.
- FT-M (Zhang et al., 2024a) Fine-tuning with masking that uses cross-entropy loss on target answers while masking original text, providing more precise weight adjustments aligned with traditional fine-tuning objectives.
- LoRA (Hu et al., 2022) Low-rank adaptation that introduces trainable rank decomposition matrices into Transformer layers, freezing pretrained weights while optimizing low-rank matrices for parameter-efficient fine-tuning.
- ROME (Meng et al., 2022) Rank-one model editing that localizes factual associations in MLP modules through causal intervention, then makes targeted rank-one parameter changes to alter individual factual associations with minimal disruption.
- GRACE (Hartvigsen et al., 2024) Sequential editing method that introduces layer adaptors with cached embeddings and codebook storage, enabling thousands of sequential edits while maintaining model stability through a deferral mechanism.
- Zero-shot Zhao et al. (2025); Zheng et al. (2023a) Zero-shot prompting that directly incorporating user preferences into the input context before presenting evaluation questions.

These model editing techniques can be categorized into the following 3 categories. We select representative editing methods (ROME, FT-M, and Zero-shot) from each category and study their effectiveness in UPQA .

- **Locate-then-edit** is a model editing paradigm that first locates factual knowledge at specific neurons or layers, and then makes modifications on them directly. We selected two typical methods ROME (Meng et al., 2022) and MEMIT (Meng et al., 2023).
- **Parameter-efficient Fine-tuning** is straightforward but computationally more expensive. We selected Fine-Tuning with Masking (FT-M) (Zhang et al., 2024a) and LoRA (Hu et al., 2022), which mitigate catastrophic forgetting and overfitting issues of standard fine-tuning.
- **In-context Editing** is a parameter-preserving paradigm that associates LLMs with in-context knowledge directly (Zheng et al., 2023a; Fei et al., 2024). We adopted a simple zero-shot baseline method in Zheng et al. (2023a) that does not provide demonstrations.

### 5.2 EVALUATION

After constructing the UPQA benchmark, we design an evaluation pipeline to assess the effectiveness of model editing methods for personalization. Our evaluation primarily follows the established model editing paradigm and uses the *Efficacy Score* (%) as the main metric. This score measures whether the edited model can generate target answers that accurately reflect user preferences, and is equivalent to the success rate. To further examine whether a personalized LLM can robustly provide preference-aware responses across diverse question types, we introduce the *Generalization Score* (%), which evaluates the model's ability to handle paraphrased or implicit questions related to the same user preference. This metric captures the percentage of personalized responses produced under more challenging conditions.

For multi-turn conversation settings, we insert inter-turn dialogue as distractions before the evaluation question. Following PrefEval Zhao et al. (2025), we retrieve these inter-turn conversational turns from the Lmsys1M dataset Zheng et al. (2023b). However, unlike PrefEval, which is designed to evaluate prompting-based methods and thus explicitly inserts user preferences into the context (structured as user preference followed by inter-turn conversation and then the evaluation question), our evaluation does not include user preference information in the context. This design more accurately reflects the personalization editing setting, where user knowledge is embedded in the model itself rather than reintroduced through prompts.

We employ Claude-4-Sonnet as the automatic judge to assess whether model responses acknowledge, reflect, or demonstrate awareness of user preferences, following prior work that defines

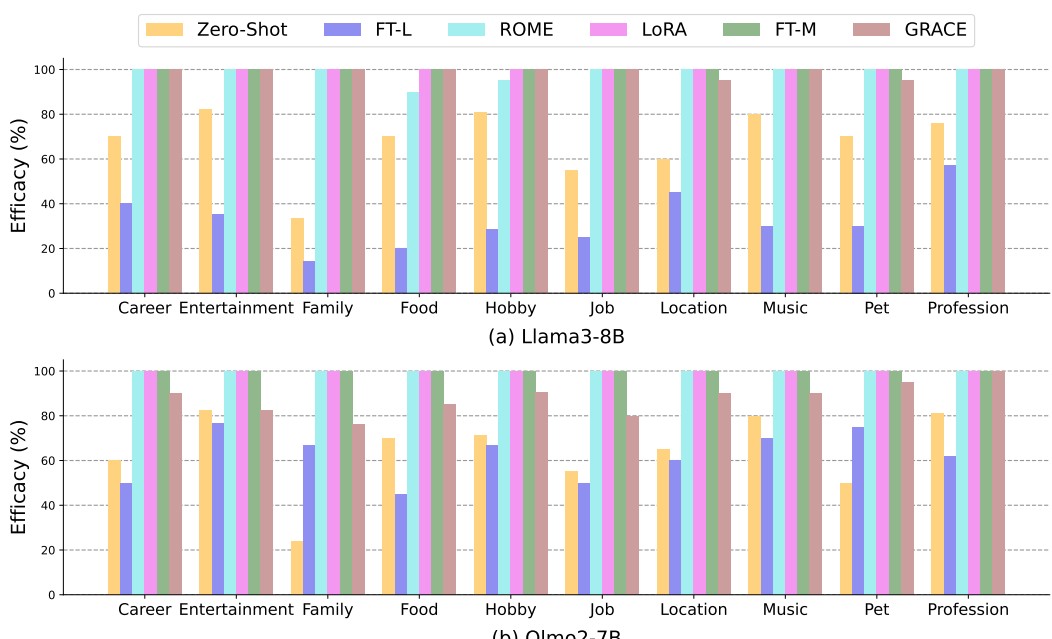

Figure 2: Evaluation results of various model-editing methods for Llama3-8B and Olmo2-7B on UPQA across 10 preference types. Efficacy scores (%) indicate the editing success rate on question–answer pairs.

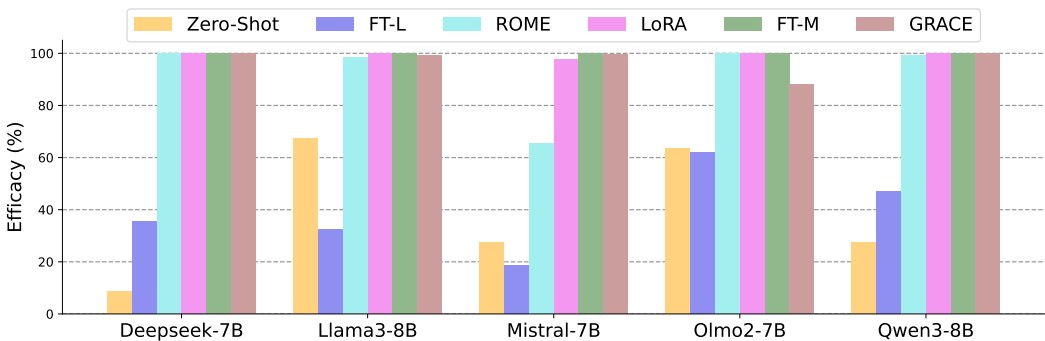

Figure 3: Efficacy score of various model editing methods across multiple LLMs on UPQA .

this metric as the *Acknowledge Rate* (%) Zhao et al. (2025). The detailed evaluation prompts used for judgment are provided in Appendix D.

## 5.3 Effectiveness of Personalization Editing

We first evaluate the effectiveness of Personalization Editing on the proposed UPQA data. Figure 2 and 3 shows that Personalization Editing consistently achieves higher Efficacy Scores across all preference types, demonstrating its ability to robustly encode user-specific information. Moreover, Figure 4 highlights that Personalization Editing generalizes effectively across six different base models.

While ROME exhibits strong efficacy on direct preference injection, it fails to generalize to rephrased questions, implicit references, and recommendation-style queries. In contrast, zero-shot prompting preserves some ability on rephrased questions but lags far behind editing-based methods in efficacy, underscoring that persistent and reliable personalization requires direct parameter updates rather than transient prompting. FT-M achieves competitive performance in generalization.

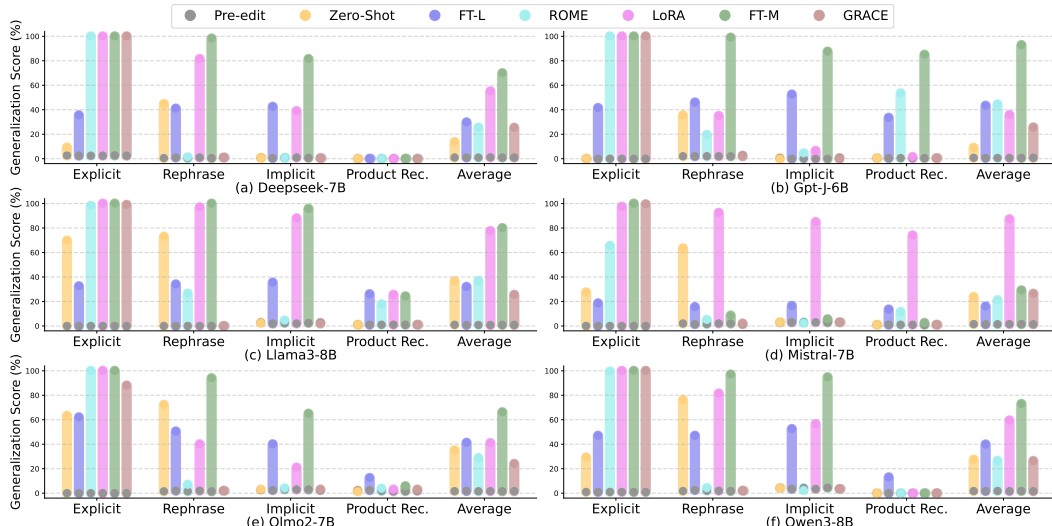

Figure 4: Generalization scores of various model editing methods across multiple LLMs on UPQA . Generalization Scores (%) are measured by accuracy on four types of Generalization Evaluation Questions including Questions ("Explicit"), Rephrased Questions ("rephrase"), Implicit Questions ("implicit"), Product-Recommendation Questions ("Product Rec."). The "Average" refers to averaged scores over four question types.

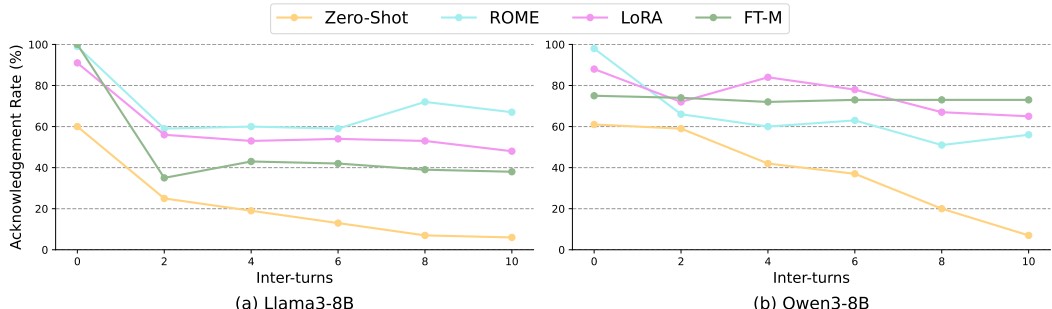

Figure 5: Acknowledgment Rate (%) over 10-turn conversations. Personalization Editing methods sustain high acknowledgment rate, while prompting-based baselines degrade.

> **Findings**
>
> **Finding 1:** Personalization Editing is highly effective at encoding user-specific facts into LLMs, enabling them to provide personalized responses aligned with user preferences.

## 5.4 SUSTAINING PERSONALIZATION BEYOND THE FIRST TURN

To evaluate whether personalization persists across extended interactions, we measure the *Acknowledgment Rate* in multi-turn dialogues on PREFEVAL (Zhao et al., 2025). As shown in Figure 5, Personalization Editing maintains high acknowledgment of user preferences throughout 10 conversational turns, demonstrating robustness even as unrelated dialogue content introduces distractions. In contrast, prompting-based methods degrade rapidly, falling below 20% by the 8th turn, as models fail to recall preferences without repeated explicit reminders. This gap highlights a key advantage of parameter-based editing: by modifying internal representations, the injected personalization becomes persistent and less susceptible to forgetting across turns, whereas prompting remains transient and fragile. These results emphasize the necessity of stable, parameter-level personalization for realistic multi-turn settings.

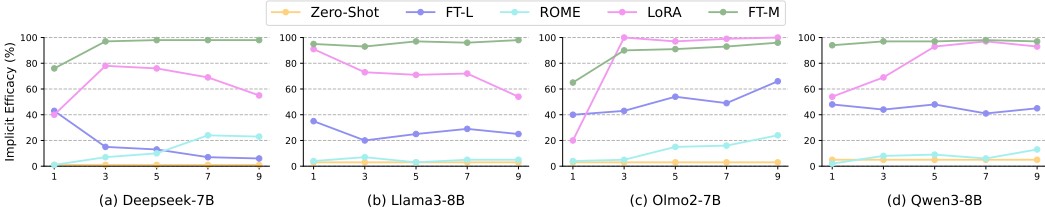

Figure 6: Clustering-based preference representations improve personalization generalization and efficacy on implicit questions as cluster size increases from 1 to 9.

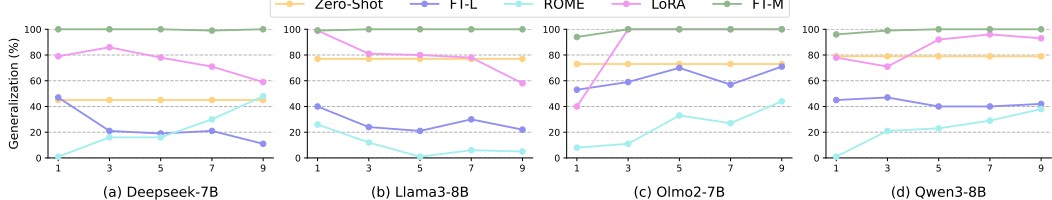

Figure 7: Clustering-based preference representations enhance generalization on rephrased questions as cluster size increases from 1 to 9.

> **Findings**
>
> **Finding 2:** Personalization Editing provides persistent personalization in multi-turn conversations, showing robustness to distractions and outperforming prompting-based methods.

## 5.5 ROBUST EDITING WITH CLUSTERING-BASED PREFERENCE REPRESENTATIONS

Real-world personalization often requires models to recall preferences that are not explicitly restated. To evaluate this setting, we focus on the implicit split of UPQA , which represents the most challenging question type. As shown in Figure 6 and Figure 7, when the cluster size is 1 (equivalent to standard model editing), editing methods already outperform zero-shot prompting. Increasing the cluster size further improves efficacy, with cluster size 3 offering a strong balance point, beyond which gains plateau. Personalization Editing, augmented with clustering-based preference representations, achieves consistently higher performance as cluster size grows. This demonstrates that clustering enables more generalizable personalization, allowing models to extend beyond literal preference mentions and adapt to rephrased or implicit formulations.

> **Findings**
>
> **Finding 3:** Personalization Editing shows strong performance for challenging implicit questions. Moreover, the clustering preference representation further improves performance on rephrased and implicit questions. Personalization Editing achieves strong performance on challenging rephrased and implicit questions, and clustering-based preference representations further improve generalization.

## 6 CONCLUSION

In this paper, we introduced *Personalization Editing*, a framework that conceptualize LLM personalization as a model-editing task, enabling precise and compute-efficient adaptation without the need for full retraining. To support rigorous and realistic evaluation, we presented UPQA , a challenging benchmark designed to directly test personalization methods on user-centric queries. Building on this formulation, we proposed a clustering-based preference representation that enhances existing editing techniques, improving accuracy, robustness, and efficiency, particularly on difficult implicit-preference queries. Extensive experiments across diverse benchmarks and model families demonstrate the effectiveness and generality of our approach, establishing Personalization Editing as a practical and versatile solution for robust LLM customization.

## 7 ETHICS STATEMENT

This research adheres to established standards of ethical conduct. No human subjects were directly involved in the study. All datasets used were either publicly available or synthetically constructed, and no personally identifiable information or sensitive user data was included. When designing the UPQA dataset, we focused on simulated user profiles to avoid privacy risks, while still capturing realistic personalization scenarios. We recognize the potential risks of personalization methods, including amplification of bias, privacy concerns, and harmful content generation. To mitigate these risks, our framework was designed to reduce hallucinations, support reversible preference editing, and avoid storing or exposing sensitive user information. In addition, the dataset and methodology are released with documentation and intended for research purposes only, with clear guidelines discouraging misuse in harmful or discriminatory applications. All experiments were conducted with transparency, and we are committed to fostering fairness, accountability, and responsible deployment of personalization technologies in real-world applications.

## 8 REPRODUCIBILITY STATEMENT

All code, data, and results are available in an anonymous repository at `https://anonymous.4open.science/r/personalization-editing`. We also provide the evaluation prompts used for LLM-Judge in Appendix D, where we specifically used the `us.anthropic.claude-sonnet-4-20250514-v1:0` model provided via AWS Bedrock. Our code additionally offers the option to run a local LLM for evaluation. We conducted all experiments on NVIDIA H200 GPUs. We recommend using a graphics card with at least 48 GB of memory. To ensure reproducibility, greedy decoding was applied across all models. The model checkpoints are downloaded from `https://huggingface.co/`. The specific download links are as follows:

- Llama-3-8B-Instruct: `https://huggingface.co/meta-llama/Meta-Llama-3-8B-Instruct`
- Mistral-7B-Instruct-v0.3: `https://huggingface.co/mistralai/Mistral-7B-Instruct-v0.3`
- Qwen3-8B: `https://huggingface.co/Qwen/Qwen3-8B`
- GPT-J-6B: `https://huggingface.co/EleutherAI/gpt-j-6b`
- DeepSeek-7B:`https://huggingface.co/deepseek-ai/DeepSeek-R1-Distill-Qwen-7B`
- OLMo-7B-Instruct-hf: `https://huggingface.co/allenai/OLMo-7B-Instruct-hf`
- Gemma2-9B-it: `https://huggingface.co/google/gemma-2-9b-it`

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

## A    THE USE OF LARGE LANGUAGE MODELS

Large language models were utilized exclusively for spell-checking and proofreading. All research ideas and the full writing of the paper were carried out entirely by the authors.

## B    TIME EFFICIENCY ANALYSIS

| Editing Method | Total File Runtime (s) | Average Edit Time (s) |
|---|---|---|
| FT-L | 3168.655000 | 1.533571 |
| FT-M | 2294.145714 | 0.221429 |
| GRACE | 3919.603333 | 4.955000 |
| ICE | 2981.761429 | 0.000000 |
| LoRA | 5208.758333 | 13.561667 |
| ROME | 3175.657143 | 2.032857 |

Table 1: Comparison of runtime and average time per edit across different editing methods on subset of UPQA with size of 200.

## C  DATA CONSTRUCTION PROMPTS

Below are the prompts for building our UPQA data.

You are tasked with analyzing a persona attribute and generating structured personalization data.

Given this persona attribute: "{persona}"

Generate a JSON response with the following fields:
1. "input_attribute": the original persona attribute (exactly as provided).
2. "attribute_type": a high-level category of the attribute (e.g., "hobby", "profession", "pet", "location", "job", "family", "food preference", "career goal"). Use the broadest appropriate category, not the specific value. Use lowercase and separate words with spaces.
3. "question": explicitly ask about the attribute type, use the exact word of the attribute_type in the question (e.g., "What's my hobby?" for a hiking-related persona).
4. "question_paraphrased": a natural rewording of the direct question.
5. "implicit_question": a conversational question that avoids directly naming the attribute type but still guides toward answers closely aligned with the target in an everyday, non-diagnostic way (e.g., "What should I do this weekend?" for a hiking hobby).
6. "product_recommendation_question": asks for a product suggestion relevant to the attribute_type without mentioning the specific attribute value (e.g., "Any gear I should buy for my hobby?").
7. "target": a concise description of what the persona reveals about the person (a single word or short phrase grounded in the input attribute).

Example input: I enjoy hiking in the mountains.
Example Output:
{
"input_attribute": "I enjoy hiking in the mountains.",
"attribute_type": "hobby",
"question": "What's my hobby?",
"question_paraphrased": "What do I like to do for fun?",
"implicit_question": "Got any suggestions for a relaxing weekend activity?",
"product_recommendation_question": "Any gear I should buy for my hobby?",
"target": "Hiking in the mountains",
}

Respond only with valid JSON, no additional text or explanation.

Below are the prompts for generating clusters of synonyms.

Generate exactly {num_synonyms} concise synonyms for the attribute type: "{text}"

Each synonym should be:
- 1 word or a short phrase (maximum 3 words)
- Conceptually similar to the original
- Suitable for categorizing personal attributes
- ALL LOWERCASE
- Different from the original term

Original: {text}

Provide only the {num_synonyms} synonyms, one per line, without numbering or bullet points. Ensure all synonyms are in lowercase.

## D   EVALUATION PROMPTS

Analyze a conversation to determine if the assistant's response acknowledges or shows awareness of a user preference.

Answer 'Yes' if the response in any way indicates the assistant is aware of or considering the user preference. Extract the part of the response that shows this awareness.

Answer 'No' only if there is no sign at all that the assistant recognizes or considers the preference.

Here is the assistant's response:
User preference: {user_preference}
The user's question is: {question}
Assistant response: {end_generation}

Reply in this exact XML format:

```
<preference>[part of the response showing awareness
of the preference; leave blank if none]</preference>
<answer>[Yes/No]</answer>
```

## E  EXAMPLES OF AUGMENTED UPQA

```
{
"input_attribute": "I love to ride my bike.",
"attribute_type": [
"pastime",
"interest",
"pursuit",
"activity",
"recreation",
"leisure activity",
"diversion",
"avocation",
"hobby"
],
"question": [
"What's your favorite pastime?",
"What's your main interest these days?",
"What's your current pursuit?",
"What's your favorite activity to do on weekends?",
"What's your favorite type of recreation?",
"What's your favorite leisure activity?",
"What's your favorite diversion when you need to unwind?",
"What's your main avocation outside of work?",
"What's my hobby?"
],
"question_paraphrased": "What do I enjoy doing in my free time?",
"implicit_question": "What's a good way to stay active and get around town?",
"product_recommendation_question": "Any gear I should buy for my hobby?",
"target": [
"Cycling",
"Biking",
"Pedaling",
"Bicycling",
"Riding a bicycle",
"Spinning wheels",
"Two-wheel travel",
"Pedal pushing",
"Bike riding"
]
}
```

