# OpenReview forum: "Towards Effective Model Editing for LLM Personalization"
_ICLR.cc/2026/Conference — ICLR 2026 Conference Withdrawn Submission_

### Official Review · Reviewer_Artt · 2025-10-28

**Soundness:** 2
**Presentation:** 2
**Contribution:** 1
**Rating:** 2
**Confidence:** 3

**Summary:**

This paper investigates the use of existing model editing techniques as a method for LLM personalization. It contrasts this approach with the common paradigms of full fine-tuning (which is resource-intensive) and in-context learning/RAG (which can be unreliable, especially in multi-turn conversations).

The paper's primary contributions are:

- A Benchmark: The introduction of UPQA (User Preference Question Answering), a new dataset for evaluating preference-following.

- An Evaluation: A comparative study of existing model editing methods (e.g., ROME, LoRA, FT-M) on this new benchmark, showing they are more robust than prompting in multi-turn dialogues.

- A Technique: A "clustering-based preference representation" is proposed, which acts as a form of data augmentation to help models generalize from explicit preferences (e.g., "I like hiking") to related implicit queries (e.g., "What should I do this weekend?").

**Strengths:**

1. The paper clearly articulates a valid and important problem: the failure of prompting/RAG-based personalization in multi-turn conversations (as demonstrated in Figure 5) and the high cost of full fine-tuning.

2. The paper conducts a thorough comparison of several representative model editing techniques (ROME, LoRA, FT-M) and baselines (Zero-Shot) on its new benchmark, providing a clear performance landscape.

3. The problem of generalizing from stated preferences to implicit user needs is a key challenge for personalization, which is a good research direction.

**Weaknesses:**

1. The paper's central concept, "Personalization Editing," does not appear to be a novel algorithm. Rather, it is a re-application of existing model editing techniques (LoRA, FT-M, ROME) to the problem of personalization. The primary technical contribution, "clustering-based preference representation," is a data augmentation strategy, not a new editing mechanism. This frames the paper more as an evaluation study than a significant methodological advancement.

2. The paper's main contribution, the UPQA dataset, has several weaknesses that question its solidity:

- "In-Situ" Claims are Misleading: The paper repeatedly describes the queries as "in-situ" (Abstract, Sec 4), which implies collection from real-world user interactions. However, the methodology (Sec 4.1, Appendix C) clearly states the dataset was synthetically generated by an LLM (Claude-Sonnet-4), which was prompted to create questions based on another synthetic dataset (Jandaghi et al., 2023). This is the opposite of "in-situ" and casts doubt on the benchmark's realism.

- Oversimplified and "Toy" Queries: The examples provided (e.g., "What's my hobby?") appear overly simplistic and directly tied to the synthetic persona attribute. It is questionable whether these queries reflect the complexity, ambiguity, and contextual richness of real user-preference queries.

- LLM-Generated Benchmark Evaluated by an LLM: There is a risk of a "self-fullfilling prophecy" when a benchmark generated by one LLM (Claude-Sonnet-4) is then evaluated by another LLM from the same family (Claude-4-Sonnet, as per Sec 5.2). This circular, synthetic methodology may not be a reliable measure of performance on genuine human-generated text.

3. A fundamental claim of any "model editing" paper is locality—that the edit does not break unrelated knowledge (i.e., it avoids catastrophic forgetting). The objective function (Sec 3.4) explicitly includes this constraint. However, the experiments (Section 5) completely lack any evaluation of locality. They measure if the edit worked (Efficacy) and if it generalized (Generalization Score), but not if it damaged the model's general capabilities (e.g., performance on a general benchmark like MMLU). This is a critical omission.

4. The experiments appear to test preferences one at a time. A real-world personalization system must handle hundreds of preferences added sequentially, including preferences that conflict or evolve over time (e.g., "I am vegan" followed later by "I love cheese pizza"). The paper provides no analysis of how "Personalization Editing" scales or handles such conflicts, which are central challenges in the field.

**Questions:**

See Weaknesses.

---

> ### Author Response · Authors · 2025-11-27
>
> Thank you for your detailed feedback. We address the specific concerns below:
> > The paper's central concept, "Personalization Editing," does not appear to be a novel algorithm. Rather, it is a re-application of existing model editing techniques (LoRA, FT-M, ROME) to the problem of personalization. The primary technical contribution, "clustering-based preference representation," is a data augmentation strategy, not a new editing mechanism. This frames the paper more as an evaluation study than a significant methodological advancement.
>
> While we use existing editing algorithms, our contribution is the framework of applying these to the personalization domain to solve specific challenges in prompting and fine-tuning (e.g., multi-turn dilution and implicit queries). Regarding "clustering-based preference representation": clusters define the editing input space $\mathcal{X}_e$​ and target space $\mathcal{Y}_e^*$​, changing which inputs are constrained by the edit rather than just adding noisy training examples. Our clustering approach is a method to enforce semantic consistency during the weight-update step, enabling the edit to generalize to implicit queries. Figures 6–7 show that this representation systematically improves robustness on implicit and paraphrased queries.
>
> > "In-Situ" Claims are Misleading: The paper repeatedly describes the queries as "in-situ" (Abstract, Sec 4), which implies collection from real-world user interactions. However, the methodology (Sec 4.1, Appendix C) clearly states the dataset was synthetically generated by an LLM (Claude-Sonnet-4), which was prompted to create questions based on another synthetic dataset (Jandaghi et al., 2023). This is the opposite of "in-situ" and casts doubt on the benchmark's realism.
>
> Our intention with “in-situ” was to contrast user-query style, short-answer questions with persona-roleplay dialogues, not to imply that queries were scraped from real-world user interactions. We used the term to describe queries generated within the context of a specific user persona, as opposed to context-less generic questions. Regarding realism: Real-world user logs are privacy-restricted and often lack ground-truth preferences. Our synthetic pipeline (anchored on validated persona attributes) allows for a controlled, efficient and reproducible benchmark. The “realism” we aim for is at the query form (short, user-centric questions) rather than source of collection.
>
> > Oversimplified and "Toy" Queries: The examples provided (e.g., "What's my hobby?") appear overly simplistic and directly tied to the synthetic persona attribute. It is questionable whether these queries reflect the complexity, ambiguity, and contextual richness of real user-preference queries.
>
> As described in Section 4, UPQA contains four graded difficulty types: direct questions, paraphrases, implicit questions, and product-recommendation queries. The examples “What’s my hobby?” etc. are from the easiest split. Concise question design with varying degree isolates the factual personalization component by removing stylistic confounds and subjectivity in evaluation. Such design requires the model to reason over the preference without explicit keywords (e.g., inferring a need for "hiking boots" from a "nature lover" persona). Our results (Fig. 5) specifically show that baselines fail on these complex queries while our method succeeds. The results in Figs. 6–7 and our “Finding 3” explicitly focus on rephrased and implicit questions, which are substantially harder, and where clustering-based editing shows the largest gains.
>
>
> > LLM-Generated Benchmark Evaluated by an LLM: There is a risk of a "self-fullfilling prophecy" when a benchmark generated by one LLM (Claude-Sonnet-4) is then evaluated by another LLM from the same family (Claude-4-Sonnet, as per Sec 5.2). This circular, synthetic methodology may not be a reliable measure of performance on genuine human-generated text.
> Because the dataset consists of short-answer QA pairs with clearly defined ground-truth labels, we employ exact string match as the primary scoring method, with LLM-as-judge (Claude-Sonnet-4) used as a fallback to handle minor variations and reduce false negatives. We manually verified scoring consistency and audited a subset of samples to ensure high evaluation fidelity. All evaluation scripts and outputs are included in the code repo for transparency and community inspection.
>
> Because the dataset consists of short-answer QA pairs with clearly defined ground-truth labels, we employ exact string match as the primary scoring method, with LLM-as-judge (Claude-Sonnet-4) used as a fallback to handle minor variations and reduce false negatives. We manually verified scoring consistency and audited a subset of samples to ensure high evaluation fidelity. All evaluation scripts and outputs are included in the code repo for transparency and community inspection.

---

> > ### Author Response · Authors · 2025-11-27
> >
> > > A fundamental claim of any "model editing" paper is locality—that the edit does not break unrelated knowledge (i.e., it avoids catastrophic forgetting). The objective function (Sec 3.4) explicitly includes this constraint. However, the experiments (Section 5) completely lack any evaluation of locality. They measure if the edit worked (Efficacy) and if it generalized (Generalization Score), but not if it damaged the model's general capabilities (e.g., performance on a general benchmark like MMLU). This is a critical omission.
> >
> > We have added additional experiments evaluating potential side effects of personalization editing. Specifically, we measure post-edit performance across two dimensions of general capabilities: (1) general knowledge (BoolQ, NaturalQuestions) and (2) reasoning ability (GSM8K, NLI). Results indicate that personalization edits introduce minimal degradation to either capability class, suggesting that the personalization method is robust and does not significantly overfit or disrupt broader model behavior.
> >
> > | Method               | GSM8K      | NLI        | BoolQ      | NaturalQuestions |
> > | -------------------- | ---------- | ---------- | ---------- | ---------------- |
> > | Pre-edit (llama3-8b) | 99.40±0.00 | 84.80±0.00 | 62.00±0.00 | 39.60±0.00       |
> > | ROME (llama3-8b)     | 99.60±0.16 | 85.00±0.28 | 61.53±1.23 | 39.73±0.38       |
> > | FT-M (llama3-8b)     | 99.47±0.09 | 85.20±0.00 | 62.13±0.09 | 39.47±0.41       |
> > | LoRA (llama3-8b)     | 99.47±0.09 | 84.07±0.50 | 61.40±1.72 | 38.40±1.30       |
> >
> > Table 1: Llama3-8b's Performance on General Knowledge and Reasoning Capacities Before and After Behavior Editing Results. Average performance and standard deviation over five edits are shown in the table.
> >
> > | Method              | GSM8K      | NLI        | BoolQ      | NaturalQuestions |
> > | ------------------- | ---------- | ---------- | ---------- | ---------------- |
> > | Pre-edit (olmo2-7b) | 99.60±0.00 | 83.20±0.00 | 58.40±0.00 | 37.27±0.34       |
> > | ROME (olmo2-7b)     | 99.53±0.09 | 83.07±0.34 | 57.27±1.11 | 35.93±0.66       |
> > | FT-M (olmo2-7b)     | 99.60±0.00 | 83.13±0.09 | 58.33±0.25 | 36.60±0.75       |
> > | LoRA (olmo2-7b)     | 99.60±0.00 | 83.67±0.25 | 58.33±0.81 | 36.33±0.09       |
> >
> > Table 2: olmo2-7b's Performance on General Knowledge and Reasoning Capacities Before and After Behavior Editing Results.
> >
> > > The experiments appear to test preferences one at a time. A real-world personalization system must handle hundreds of preferences added sequentially, including preferences that conflict or evolve over time (e.g., "I am vegan" followed later by "I love cheese pizza"). The paper provides no analysis of how "Personalization Editing" scales or handles such conflicts, which are central challenges in the field.
> >
> > Our current work deliberately focuses on single-preference and small-edit personalization, because firstly, it allows us to isolate the effect of the personalization edit and the clustering representation without conflating it with complex preference aggregation or conflict resolution. Secondly, it aligns with much of the existing model editing literature [1, 2], which primarily analyzes single edits before moving to large-scale multi-edit methods.
> > Thus, this paper focuses on the feasibility of editing for personalization (solving the "dilution" problem in RAG and the "implicit" problem). Handling conflicting updates is a general open problem in the Model Editing field rather than a specific flaw of our Personalization framework. We view this as a separate scope but will add a discussion on conflict resolution in the limitations section. We see our contribution as a first step that establishes the viability and benefits of editing-based personalization, with scaling to hundreds of evolving preferences as a promising extension.
> >
> > [1] Zhang, Ningyu, et al. "A comprehensive study of knowledge editing for large language models." arXiv preprint arXiv:2401.01286 (2024).
> >
> > [2] Wang, Song, et al. "Knowledge editing for large language models: A survey." ACM Computing Surveys 57.3 (2024): 1-37.

---

### Official Review · Reviewer_Bwcd · 2025-10-30

**Soundness:** 2
**Presentation:** 3
**Contribution:** 2
**Rating:** 4
**Confidence:** 3

**Summary:**

This work frames personalization as model editing, proposing Personalization Editing with clustering-based preference representations and introducing UPQA, a short-answer benchmark. It achieves efficient, robust personalization than prompting/fine-tuning across multiple LLMs.

**Strengths:**

+ This work frames personalization as model editing, enabling efficient and persistent parameter-level updates that outperform prompting in multi-turn and implicit-query settings.
+ This work introduces UPQA, a realistic short-answer benchmark with explicit, rephrased, implicit, and recommendation queries plus synonym clusters for robust evaluation.
+ This work proposes clustering-based preference representations that generalize to paraphrases and implicit requests, improving accuracy and multi-model robustness.

**Weaknesses:**

- Personalization metric relies on short-answer accuracy, failing to capture nuanced preferences, trade-offs, or long-term satisfaction across diverse tasks and interactions.
- No human evaluation prevents measuring perceived relevance, satisfaction, or harms from personalized edits in real users.
- Focus on short answers limits applicability to open-ended tasks, multi-turn dialogues, or creative personalization needs—unclear generalization to longer responses.
- Parameter edits may cause unintended behavior, overfitting, or harmful side effects; robustness, rollback, and safety under adversarial preferences are underexplored.

**Questions:**

Please see Weaknesses.

---

> ### Author Response · Authors · 2025-11-26
>
> We appreciate your feedback and please find detailed responses below.
>
> > Personalization metric relies on short-answer accuracy, failing to capture nuanced preferences, trade-offs, or long-term satisfaction across diverse tasks and interactions.
>
> Short-answer QA is intentionally chosen as a grounded and controlled setting to evaluate whether models can correctly recall user-specific information. Short-answer QA isolates the factual personalization component by removing stylistic confounds and subjectivity in evaluation. This format provides clear correctness criteria and minimizes ambiguity during scoring. While alternative evaluation formats, such as personalized text generation or stylistic imitation [1], can offer richer signal, they are more subjective, costlier to evaluate at scale, and prone to noise or evaluator disagreement. Our use of short-answer accuracy therefore serves as a precise and scalable first-order measure of factual personalization performance.
>
> > No human evaluation prevents measuring perceived relevance, satisfaction, or harms from personalized edits in real users.
>
> Because the dataset consists of short-answer QA pairs with clearly defined ground-truth labels, we employ exact string match as the primary scoring method, with LLM-as-judge (Claude-Sonnet-4) used as a fallback to handle minor variations and reduce false negatives. We manually verified scoring consistency and audited a subset of samples to ensure high evaluation fidelity. All evaluation scripts and outputs are included in the code repo for transparency and community inspection.
>
> > Focus on short answers limits applicability to open-ended tasks, multi-turn dialogues, or creative personalization needs—unclear generalization to longer responses.
>
> Short-answer tasks cannot capture the full space of personalization scenarios, particularly those involving rich generation or interaction. However, this design offers a cost-efficient and standardized way to benchmark personalization quality at scale, especially for factual recall, an ability foundational to downstream personalized text generation. The dataset includes both explicit and implicit personalization cues, reflecting a wide range of real-world human–AI interactions. Additionally, we include multi-turn personalization experiments in Section 5.4 to demonstrate generalization beyond isolated short-answer responses. If possible, we would appreciate clarification on the term "creative personalization needs,". Finally, short-form responses also reduce token overhead, enabling rapid experimentation and reproducible comparison.
>
> > Parameter edits may cause unintended behavior, overfitting, or harmful side effects; robustness, rollback, and safety under adversarial preferences are underexplored.
>
> We have added additional experiments evaluating potential side effects of personalization editing. Specifically, we measure post-edit performance across two dimensions of general capabilities: (1) general knowledge (BoolQ, NaturalQuestions) and (2) reasoning ability (GSM8K, NLI). Results indicate that personalization edits introduce minimal degradation to either capability class, suggesting that the personalization method is robust and does not significantly overfit or disrupt broader model behavior.
>
> | Method               | GSM8K      | NLI        | BoolQ      | NaturalQuestions |
> | -------------------- | ---------- | ---------- | ---------- | ---------------- |
> | Pre-edit (llama3-8b) | 99.40±0.00 | 84.80±0.00 | 62.00±0.00 | 39.60±0.00       |
> | ROME (llama3-8b)     | 99.60±0.16 | 85.00±0.28 | 61.53±1.23 | 39.73±0.38       |
> | FT-M (llama3-8b)     | 99.47±0.09 | 85.20±0.00 | 62.13±0.09 | 39.47±0.41       |
> | LoRA (llama3-8b)     | 99.47±0.09 | 84.07±0.50 | 61.40±1.72 | 38.40±1.30       |
>
> Table 1: Llama3-8b's Performance on General Knowledge and Reasoning Capacities Before and After Behavior Editing Results. Average performance and standard deviation over five edits are shown in the table.
>
> | Method              | GSM8K      | NLI        | BoolQ      | NaturalQuestions |
> | ------------------- | ---------- | ---------- | ---------- | ---------------- |
> | Pre-edit (olmo2-7b) | 99.60±0.00 | 83.20±0.00 | 58.40±0.00 | 37.27±0.34       |
> | ROME (olmo2-7b)     | 99.53±0.09 | 83.07±0.34 | 57.27±1.11 | 35.93±0.66       |
> | FT-M (olmo2-7b)     | 99.60±0.00 | 83.13±0.09 | 58.33±0.25 | 36.60±0.75       |
> | LoRA (olmo2-7b)     | 99.60±0.00 | 83.67±0.25 | 58.33±0.81 | 36.33±0.09       |
>
> Table 2: olmo2-7b's Performance on General Knowledge and Reasoning Capacities Before and After Behavior Editing Results.
>
> [1] Salemi, Alireza, et al. "Lamp: When large language models meet personalization." Proceedings of the 62nd Annual Meeting of the Association for Computational Linguistics (Volume 1: Long Papers). 2024.

---

### Official Review · Reviewer_G3jE · 2025-11-01

**Soundness:** 3
**Presentation:** 2
**Contribution:** 2
**Rating:** 2
**Confidence:** 3

**Summary:**

The authors proposed a novel Personalization Editing framework to better reflect user demand. The authors also introduced a short-answer UPQA for evaluating Personalization Editing. The proposed framework demonstrates strong and robust editing performance.

**Strengths:**

* The proposed Personalization Editing is an interesting and novel editing task.
* The proposed framework demonstrated and robust strong performance on editing

**Weaknesses:**

* The formatting in Section 3 looks a bit disorganized with few sentences in a subsection, I recommend that the authors summarize the problem setups in fewer subsections and elaborate a bit more on the objective function, say, what exactly is the loss function $\mathcal{L}$?

* The overall novelty of this work is limited, the theoretical and empirical insight may not benefit broader audience beyond this task.

**Questions:**

Please see Weaknesses

---

> ### Author Response · Authors · 2025-11-26
>
> > The formatting in Section 3 looks a bit disorganized with few sentences in a subsection, I recommend that the authors summarize the problem setups in fewer subsections and elaborate a bit more on the objective function, say, what exactly is the loss function $\mathcal{L}$?
>
> We thank the reviewer for pointing out the structural issues in Section 3. In the revision, we will consolidate the original Subsections 3.1, 3.2, and 3.3 into a single subsection titled 3.1 Problem Formulation. This will provide a more cohesive reading flow regarding the input-output mapping and personalization definitions. Regarding the objective function in the original Section 3.4, we will clarify that $\mathcal{L}$ represents the standard Cross-Entropy loss calculated over the target personalized response tokens, conditioned on the input query. We will update the text to explicitly define this:"...The optimization aims to minimize the discrepancy between the personalized output and the desired target output.
>
> > The overall novelty of this work is limited, the theoretical and empirical insight may not benefit broader audience beyond this task.
>
> We respectfully disagree that the novelty is limited. We believe this work offers specific technical and theoretical contributions that extend beyond the immediate task of personalization and address fundamental challenges in adaptation, continual learning, controllability, and grounding in LLMs:
>
> 1.  Novelty of the Editing Framework:
> While standard model editing focuses on rigid fact retrieval (e.g., $(s, r, o) \rightarrow (s, r, o^*)$), we identify a theoretical gap: standard editing fails when the trigger is implicit or semantically varied.
> We introduce Clustering-based Preference Representation, a novel mechanism that aggregates semantic clusters (Section 4.1) to guide edits. This addresses a core limitation in the general Model Editing field, the "Generalization" problem. As shown in Figure 6 and Figure 7, our method allows edits to generalize to implicit reasoning questions (e.g., “What should I do this weekend?” from a Hiking preference). This mechanism is not tied to the specific dataset or task, and could be extended to other tasks.
>
> 2. Empirical Insights With Broader Implications (Multi-turn Robustness):
> We provide empirical evidence (Section 5.4, Figure 4) that In-Context Learning (ICL) suffers from "context dilution" and forgetting in multi-turn dialogues (degrading to <20% acknowledgment rate), whereas Personalization Editing remains stable.
> This offers a crucial insight for researchers working on Long-Context LLMs and Memory: parameter updates are necessary for persistent behavior in noisy, multi-turn environments where context windows become unreliable These insights extend beyond personalization to any setting where a model must retain and apply localized user- or task-specific knowledge over time.
>
> 3. New Benchmarking Resource for the Community:
> Existing personalization datasets evaluate style, persona imitation, or role-play rather than grounded user-specific memory. Our proposed UPQA dataset fills this gap and enables standardized model editing evaluation on implicit questions and multi-turn settings. This provides the broader community with a standardized testbed for evaluating Preference Retrieval and Reasoning capabilities in personalized settings.

---

### Official Review · Reviewer_g3aB · 2025-11-07

**Soundness:** 2
**Presentation:** 3
**Contribution:** 3
**Rating:** 2
**Confidence:** 3

**Summary:**

The paper introduces Personalization Editing, which novelly frames the LLM personalization problem as a model editing task. It then proposes clustering-based preference representation and uses it to extend existing model editing methods. The paper also presents a new dataset, UPQA, which can directly test personalization methods on user queries of various difficulty levels.

Experiments show that model editing methods constantly outperform the zero-shot method, where user preference is incorporated directly into the model prompt. As the number of turns increases, the model editing methods are also more robust than the zero-shot method. Besides, the clustering-based preference representation improves some of the editing methods.

**Strengths:**

1. The paper proposes a novel framework that conceptualizes LLM personalization as a model-editing task, so that the existing model-editing methods can be adopted to tackle the LLM personalization problem.

2. The paper presents a new dataset, UPQA, together with clearly defined metrics, which can serve as a new benchmark for LLM personalization methods.

3. The paper extensively experiments with existing model-editing methods and presents comprehensive experimental results.

**Weaknesses:**

1. Although the paper presents definitions and high-level formulas for the Personalization Editing framework, such as input, output, objective, etc., it lacks sufficient details about how existing model-editing methods are used to be compatible with the framework and the evaluation datasets. The constrained objective is conceptually clear, but integration details per editor (e.g., layer selection, masking strategy) are not specified.

2. The paper primarily augments existing editors, and no new low-level weight-update algorithm is introduced, which limits its originality.

3. The paper uses Claude as the judge. While it shares the evaluation prompt, a validation of the prompt (e.g., human validation) is missing.

**Questions:**

1. For the Experiments section, you are comparing a list of model-editing methods to the zero-shot baseline. Could you clarify what changes you made to these model-editing methods? Did you integrate any innovative improvements to them, or use them off-the-shelf?

2. In Figures 2 and 3, we see that a few methods (such as LoRA and FT-M) already achieve 100% on almost every model and preference type.  Does this indicate that the proposed dataset (UPQA) is not difficult / diverse enough?

3. As the Weakness part mentioned, could you share more implementation-level details about per-editor integration and clustering integration?

4. Figure 6 shows that increasing cluster size negatively affects FT-L's performance on Deepseek-7B and Llama3-8B. Could you elaborate more on the effectiveness and generalizability of the clustering method?

5. Two anonymous repo links are provided, but one is empty (only contains a README doc), and the other one has expired. Could you re-share the anonymous repo if available?

---

### Note · Authors · 2025-12-29

I have read and agree with the venue's withdrawal policy on behalf of myself and my co-authors.